# Microbial Contamination and Disease Outbreaks Associated with Rockmelons (*Cucumis melo*): Implications for Public Health Protection

**DOI:** 10.3390/foods13142198

**Published:** 2024-07-11

**Authors:** Pouria Rabiee, Ayesha Faraz, Said Ajlouni, Malik A. Hussain

**Affiliations:** 1School of Science, Western Sydney University, Hawkesbury Campus, Richmond, NSW 2753, Australiaa.faraz2@westernsydney.edu.au (A.F.); 2School of Agriculture, Food, and Ecosystems Sciences, Faculty of Sciences, The University of Melbourne, Parkville, VIC 3010, Australia; said@unimelb.edu.au

**Keywords:** cantaloupe, *Listeria monocytogenes*, *Salmonella*, *Escherichia Coli*, foodborne illness, food poisoning

## Abstract

Foodborne illnesses caused by consuming contaminated fresh produce not only pose serious public health risks but also lead to huge economic losses. Rockmelons (cantaloupes) have emerged as a recurrent source of disease outbreaks caused by foodborne pathogens, including *Listeria monocytogenes*, *Salmonella*, and *Escherichia coli*. The most common factor of the outbreaks was the microbial contamination of rockmelons at the farm, and subsequently, the pathogenic bacteria were transferred to the flesh during cutting and processing. One of the deadliest outbreaks occurred in the USA due to *L. monocytogenes* contamination of rockmelons which caused 33 deaths in 2011. Since then, several guidelines and recommendations have been developed for food safety management to reduce the microbial contamination of melons on farms and post-harvest operations. This article explicitly provides an updated overview of microbiological contamination, disease outbreaks, pathogens prevalence, and mitigation strategies to reduce public health risks due to the consumption of rockmelons.

## 1. Introduction

Disease outbreaks, foodborne illnesses, product recalls, and other food safety incidents continue to be linked to consumption of contaminated fresh produce. Fresh produce safety not only impacts public health but also the economy in a substantial manner. Along with the public health burden, fresh produce contamination and disease outbreaks inflict real economic losses on the food businesses and industry [1]. Fresh fruits and vegetables are a critical part of a balanced diet and provide essential nutrients and vitamins. However, their susceptibility to microbial contamination causes great concerns to the food industry and regulatory bodies for improving the microbiological safety of fresh produce [2]. With the emphasis on healthy eating and increased fresh produce consumption, producers are under more pressure to focus on control measures to minimise the prevalence of food safety hazards. This is especially relevant for the produce grown in open fields, which could be exposed to various physical, chemical, and biological contaminants. The most alarming situation occurs when farmers and consumers fail to wash off contaminants from the produce before consumption [3].

Fresh produce can inadvertently harbour harmful microorganisms (pathogens), including bacteria, viruses, and parasites, that are capable of causing serious health issues in human beings. Pathogens commonly associated with fresh produce include *Aeromonas* species (spp.), *Bacillus cereus*, *Campylobacter* spp., *Clostridium botulinum*, *E. coli*, *L. monocytogenes*, *Salmonella enterica, Shigella* spp., *Staphylococcus* spp., *Yersinia enterocolitica*, Hepatitis A, norovirus, and *Cyclospora* [4]. The consumption of raw or minimally processed fresh produce could expose consumers to potential health risks due to the presence of pathogenic microorganisms. Additionally, the abundant water content within fresh produce provides an environment conducive to the proliferation of microorganisms. There is an increasing need to assess, understand, and alleviate the microbial contamination risks associated with fresh produce [5].

The rockmelon is a type of true melon (*Cucumis melo*) that belongs to the family *Cucurbitaceae*. It is a netted muskmelon called ‘cantaloupe’ in North America and ‘rockmelon’ in Australia and New Zealand. Melons are low-calorie fruits with high water content and are considered a good source of potassium, vitamin B, and vitamin C [6]. The melon industry in Australia holds an annual value of around $172 million [7]. This industry involves about 140 growers who cultivate a variety of melons, including watermelons, rockmelons, honeydews, and specialty types, covering approximately 8500 hectares of land [7]. Melon production occurs consistently throughout the year. While melons are cultivated in numerous states and territories across Australia, the primary regions for cultivation are Cowra, Riverina, and Sunraysia in New South Wales [7,8].

Rockmelons have been linked to several serious foodborne illness outbreaks in recent years; however, the deadliest *Listeria* outbreak occurred in the USA and caused 33 deaths in 2011 [9]. This article provides an updated overview of the microbiological contamination, disease outbreaks, pathogens prevalence, and food safety management of rockmelons.

## 2. Microbial Contamination of Rockmelons

Rockmelons are susceptible to microbial contamination from various sources at different stages within the supply chain [10]. Figure 1 highlights some common sources and modes of cross-contamination in rockmelons in the supply chain. One of the primary sources of concern is the environment in which they are grown [11]. Interaction with contaminated soil and irrigation water can introduce pathogens to the surface of the rockmelons [12]. Additional factors such as wild animals, pests, and insects can contribute to microbial contamination of the growing environments. Wild animals and pests can introduce pathogens to the environment and potentially onto the rockmelons. Human activities constitute another pivotal source of contaminants due to poor hygiene and handwashing practices at various stages of rockmelon production. When not properly washed, workers’ hands can carry harmful microorganisms that could be transferred to the rockmelons during harvesting, sorting, or packaging. Moreover, cross-contamination through equipment, surfaces, or tools that are not effectively cleaned and sanitised can lead to the accidental spread of contaminants throughout the rockmelon crop [13].

Transportation and storage stages within the supply chain are critical points where contamination can occur. During transportation, inadequate temperature control can lead to the growth of existing microorganisms on the rockmelons. Cross-contamination is also a concern if transport vehicles are not thoroughly cleaned between shipments, allowing pathogens from previous lots to contaminate subsequent lots. Similarly, storage facilities play a role in the growth of contaminating microorganisms. For example, fluctuations in storage temperatures could promote the proliferation of pathogens, and insufficient sanitation practices would enhance the chances of cross-contamination.

The retail environment and consumer handling could also cause contamination. Improper washing of rockmelons before consumption can transfer contaminants from the surface to the edible portions when the fruit is cut. Retail display areas not regularly cleaned and sanitised could spread pathogens to the rockmelons. In contrast, consumers who handle the fruit without clean hands can enhance the risk of microbial cross-contamination. Packaging materials used for rockmelons can also be both a source and a mode of contamination if they are not adequately sanitised or are exposed to unsanitary conditions [7,14].

Understanding these diverse contamination sources and modes of cross-contamination is vital for developing effective food safety control measures in the production and distribution of rockmelons. Implementing stringent hygiene practices, regular equipment and facility sanitation, proper temperature control, and consumer education are all integral components to narrowing the risk of contamination and establishing the integrity of the rockmelon supply chain. However, failure to manage microbial contamination could become a public health protection concern especially if the microorganisms are pathogenic and capable of causing foodborne illnesses [15].

## 3. Disease Outbreaks Associated with Rockmelons

Contaminated rockmelons were linked to numerous disease outbreaks around the world. Some notable disease outbreaks associated with rockmelons are listed in Table 1. It is noted that *L. monocytogenes* and *S. enterica* (different serovars) have been the two main foodborne pathogens attributed to most disease outbreaks linked to rockmelon consumption since 2006. The primary challenge faced by the global melon industry, particularly rockmelons, is the pervasive issue of microbial contamination. The continuous occurrence of major outbreaks associated with rockmelons every few years worldwide provides the difficult nature of this challenging situation. Bowen et al. [16] summarised the infections associated with rockmelon consumption including 1434 illnesses, 42 hospitalisations, and 2 deaths from twenty-three outbreaks that occurred between 1984 and 2002. This rockmelon safety challenge is equally significant in Australia. In 2006, *S. enterica* serovar Saintpaul contamination-linked outbreak caused 100 illnesses and 9 hospitalisations in Australia [17]. The susceptibility of rockmelons to microbial contamination is because of close contact with soil during their final growth phase and the coarse net-like outer skin. Among the various microorganisms, *Salmonella* spp. and *L. monocytogenes* stand out as the prevalent culprits responsible for causing foodborne diseases in humans who have consumed contaminated rockmelons [18]. *E. coli* has also been isolated from rockmelons [19]. *E. coli*, encompassing variants like *E. coli* O157:H7, can trigger gastrointestinal problems if rockmelons carry such contaminants [20]. Two major outbreaks linked to rockmelons occurred in Australia in 2016 (*Salmonella*) and 2018 (*Listeria*) [21,22]. In 2023, *Salmonellosis* outbreaks linked to rockmelons were reported in the United States and Canada [23,24].

In 2011, the deadliest *Listeria* outbreak in the USA that led to 147 cases of listeriosis including 33 deaths across 28 US states was linked to consumption of *L. monocytogenes* contaminated rockmelon [9]. The disease outbreak occurred due to the contamination of whole rockmelons on a single farm, subsequent investigation identified several risk factors such as inadequate facility and equipment designs, unsanitary conditions and processing facilities, and poor hygiene practices. During the 2018 Australian *Listeria* outbreak, *L. monocytogenes* was detected on whole and cut (half) rockmelons causing 22 cases of listeriosis including 7 deaths across four states [7,21]. In that case, the contaminated rockmelons were also traced back to a single farm in New South Wales, and the main identified risk factor was heavy rain followed by dust storms before the harvest [21].

*Salmonella* outbreaks from rockmelons are frequent and more in numbers (Table 1) and are likely to arise due to contamination in the field from irrigation water, extreme weather events, soil, and animal faeces. The 2016 *Salmonella* outbreak in Australia was linked to general hygiene of the facility (excessive dirt and dust), inadequately cleaned recirculated water, and lack of appropriate monitoring of chlorine-based sanitiser in the wash water [22]. Last year, rockmelons contaminated by *Salmonella* sickened more than 400 people and caused 158 hospitalisations and 6 deaths across 44 states in the USA. The first cases of salmonellosis were confirmed in October 2023. Following this *Salmonella* outbreak, the FDA announced the first voluntary recall in November 2023, and later, 11 other companies also issued voluntary recalls.

A recall is a vital post-outbreak response after identifying a product as the source of contamination in an outbreak. The recall process involves removing or correcting the affected product from the market to prevent further illnesses. There are two distinct categories of food recalls: named consumer and trade. A trade recall is initiated when the food product has not been directly accessible to the general public and has primarily been distributed to wholesalers and caterers. Conversely, a consumer recall is enacted when the food product has been made available for retail sale to the public. Over the period spanning from 2013 to 2022, consumer recalls constituted the majority, making up 87% of all food recalls, while trade recalls accounted for the remaining 13% [22]. Notable recalls associated with microbiological contamination of rockmelons are listed in Table 2.

As a consequence of disease outbreaks, food business operators could face many setbacks to business activity in the form of increased inspections, recall actions, and delisting by retailers. Post-outbreak procedures occur once the investigation into the outbreak reaches its conclusion. In addition to determining that the outbreak has ended, several concluding activities must be carried out. These include the creation of a final epidemiological summary, communicating with relevant public health partners who were affected, and disseminating outbreak findings through knowledge-sharing efforts and tools such as outbreak summaries [29].

## 4. Survival and Persistence of Pathogenic Microorganisms

Survival and persistence of microorganisms in fresh produce such as rockmelon involve the ability of microorganisms to endure and remain viable on or within the fruits. Various factors, such as the texture of the fruits’ surface, moisture content, temperature, and storage conditions, determine how long a microorganism can persist on or inside the fruit. The survival of microorganisms on the surface or inside the pulp of rockmelons is influenced by various factors including pH, temperature, moisture, and surface condition. The handling and cross-contamination of rockmelons throughout the supply chain are critical, as initial microbial communities on the fruit surface would influence which pathogen outcompetes the others. The packaging materials and storage period also impact the prevalence of different pathogens [15].

The rough and netted surface of rockmelon provides an ideal environment for *L. monocytogenes* to thrive. Within the crevices and recesses of the rind, *Listeria* firmly adheres and forms biofilms, protective communities of microorganisms encased in a self-produced matrix. Improper cleaning and sanitising programs or moist areas, not exposed to sanitisers, can harbour *Listeria* spp. and form a biofilm, making it harder to remove [30]. Weis and Seeliger [31] reported a 20% prevalence of *L. monocytogenes* in soil. It can grow in a wide range of temperatures (0–45 °C) and is capable of surviving for extended periods in the environment [32], making it more difficult to manage than other foodborne pathogens such as *E. coli* and *Salmonella* [33]. *Listeria*’s adaptability to a wide temperature range is a concern when it comes to the storage of rockmelons. Even refrigeration, a common practice to extend produce shelf life, may only slow *Listeria* growth but not completely halt it, since *Listeria* is a psychrophilic microorganism. This is troubling as many consumers refrigerate rockmelons, potentially leading to the bacterium’s persistence, especially if the fruit is not thoroughly washed and *Listeria* is transferred to the flesh during cutting [34].

*Salmonella*, another concerning pathogen, can persist and thrive on rockmelons, posing a significant risk to consumers. Like *Listeria*, the fruit surface characteristics are pivotal in *Salmonella* survival. The rough and netted exterior provides shelter and attachment points for the bacteria, enabling them to endure and potentially flourish [35]. *Salmonella* can persist on undamaged rockmelon rind for up to 14 days, and it can also thrive in the injuries or wounds on the surface of the rockmelon. *Salmonella* spp. exhibit remarkable adaptability to a broad range of temperatures, similar to *Listeria*. This resilience means that refrigeration might not be effective in preventing *Salmonella* persistence [36].

*E. coli*, specifically the pathogenic strain *E. coli* O157:H7, is notorious for its association with foodborne outbreaks, and several crucial factors influence its survival on rockmelons. The textured exterior of rockmelons, characterised by a rough surface, plays a pivotal role in *E. coli* persistence. This roughness traps *E. coli*, making it highly resilient and resistant to elimination during cleaning and sanitation procedures. The fruit’s irregular surface provides protected niches where the bacterium can securely adhere, defying conventional disinfection methods [16]. *E. coli* O157:H7 displays remarkable adaptability to a range of temperatures, including lower ones. In the case of *E. coli* O157:H7, the number of bacteria present on the intact rockmelon rind can multiply significantly, increasing by two orders of magnitude within just 4 days at a temperature of 25 °C. This adaptability poses a significant challenge for effective pathogen control, allowing it to persist and potentially even grow on the fruit surface, particularly in refrigeration conditions [16].

Moisture content on the rind also contributes to the survival of different microorganisms including *Listeria*, *E. coli*, and *Salmonella* spp. A moist rind creates a conducive environment for a bacterium to persist and multiply. *Listeria* demonstrates remarkable versatility, capable of persisting and growing in both acidic (pH 5.0) and alkaline (pH 9.0) conditions, making it a highly adaptable pathogen. Even after washing rockmelons, if the surface retains moisture or is not adequately dried, *Listeria* can continue to survive and potentially multiply. Therefore, thorough drying is crucial to reduce moisture that supports *Listeria* persistence [37]. *Salmonella*, which can endure slightly alkaline conditions, exhibits optimal growth between pH 7.0 and 8.0. Even after rigorous washing, these moist conditions create an environment conducive to the bacterium’s persistence and potential growth [38]. Several investigations that have reported the occurrence of microbiological hazards in rockmelons are summarised in Table 3.

Effective cleaning and sanitisation protocols during processing and handling, including thorough washing, rinsing, and use of appropriate sanitiser at the correct concentration, are essential to reduce the prevalence of pathogens on rockmelons. A holistic understanding of these factors is crucial for mitigating the risks associated with *Listeria*, *Salmonella*, and *E. coli*, in rockmelons, and ensuring food safety throughout the supply chain [46].

## 5. Mitigation Strategies to Reduce Microbial Contamination of Rockmelons

The most common situation in disease outbreaks linked to rockmelons involves contamination at the farm, with the risk of contamination being amplified during the cutting process and subsequent exposure to improper temperatures. To address this issue, measures to control a combination of pre-harvest and post-harvest should be implemented to minimise the chances of contamination during the production, transportation, storage, and consumer handling of rockmelons [47]. Specifically tailored Good Agricultural Practices (GAPs) for rockmelon cultivation should be adopted, covering pre-harvest measures such as ensuring water quality, promoting hygienic practices among workers, managing manure used as fertiliser, preventing faecal contamination from animals and humans, and maintaining cleanliness in facilities and equipment. In the post-harvest stage, attention should be given to using suitable water for processing, ensuring worker hygiene, and practising proper sanitation of packing equipment and facilities along with consumer education about food safety practices at home [46]. A comprehensive approach is essential to consistently reduce the presence of pathogens at every stage, from production to consumption of rockmelons. Nevertheless, developing more effective measures necessitates a deeper understanding of how pathogens become attached and grow on melons. Research in these areas is crucial for devising strategies to minimise the risk of diseases associated with melon consumption [8].

To reduce the risk of rockmelon contamination with *L. monocytogenes* and *Salmonella* spp., it is essential to implement robust control measures during both the initial production and subsequent processing stages. This entails the implementation of GAPs at the farm level, the adherence to Good Hygienic Practices (GHPs) throughout the entire supply chain, the application of Good Manufacturing Practices (GMPs) during the processing phase, and the rigorous control of inputs throughout the supply chain [46,47]. Considering that *Listeria* is commonly present in soil, rockmelons grown on the ground face a higher risk of *Listeria* contamination compared to those cultivated above ground. Furthermore, rockmelons cultivated on the ground are at an elevated susceptibility to *Salmonella* contamination compared to those grown in a suspended manner above the ground.

Other factors during the initial melon production stage that contribute to risk include the quality of irrigation water, the use of water-soluble agricultural chemicals, the application of untreated or inadequately treated manure as fertiliser, the potential for animal intrusion, and environmental variables like site location and extreme weather events such as dust storms, heavy rainfall, and floods. To manage these risks effectively, it is important to implement GAPs that include using water of appropriate quality, such as clean or potable water when applying agricultural chemicals and for direct irrigation. It also minimises rockmelon contact with soil, soil additives, and irrigation water, which can be achieved through sub-surface or drip irrigation methods rather than overhead irrigation. Properly managing fertiliser storage and treatment facilities, understanding the previous use of the land, limiting wildlife access to the growing area, and employing windbreaks to create a protective barrier against adverse weather conditions are all strategies to mitigate risk during melon cultivation [16,18,48,49]. *Salmonella* is frequently detected in surface water used for irrigation. Trace-back investigations of outbreaks often implicate irrigation water as a source (or a vehicle) for transmission of *Salmonella.* Regular testing and treatment of irrigation water sources can help ensure its safety. Drip irrigation systems that minimise contact with the edible parts of the plant can also be beneficial [50].

In Australia, the NSW Department of Primary Industry (NSW DPI) developed and released a set of industry recommendations to improve rockmelon food safety practices in 2019 [18]. These recommended practices should be implemented along with established programs such as GAPs and hazard analysis critical control points (HACCP) [46]. The primary purpose of these guidelines is to ensure food safety practices in the supply chain for identifying, assessing, and managing potential food safety hazards associated with rockmelon at pre- and post-harvest stages. These recommendations include the use of drinking quality water for washing, pretreatment of wash water with a sanitiser, guidelines on monitoring and recording sanitiser concentration, cleaning and sanitisation of equipment, packaging line and packing house, personal hygiene, and pre-cool and storage conditions. Applications of sanitisers in post-harvest operations appear to be most efficient when employed to decrease microbial counts within the wash water, aiding in the prevention of bacterial infiltration and the potential for cross-contamination through the wash water. However, these treatments alone are not sufficient for pathogen elimination during the processing phase [51]. In post-harvest operations for rockmelons washing and sanitisation serve as a critical process to reduce the microbial load of the fruit surface. Therefore, the validation of the effectiveness of washing and sanitisation processes is crucial to ensure the safety status of the product.

Lastly, maintaining and regularly updating the standard operating procedures (SOPs), for all operations of rockmelon production and supply, is an important part of the risk mitigation strategy. An SOP is a documented set of instructions describing a specific procedure or task. SOPs provide essential information identifying the requirement, the responsible person, and how exactly to perform the procedure. SOPs are generally operation-specific and could differ from farm to farm but they ensure that the same task is completed consistently and safely by any individual operating the system.

Bartlett et al. [7] developed general recommendations to reduce contamination of pathogenic microorganisms, using five best practice guides specific for melon production. Briefly, key recommendations include the following:Pre-harvest risk management examples:Domestic animals, wildlife, insects, and pests should be controlled, reduced or eliminated.In case evidence of animal damage or faecal contamination is found, harvesting should be postponed.Acceptability of water quality depends on the intended use of the water.Harvest risk management examples:During harvest risks identified are related to equipment cleaning, people, transportation, and mechanical damage.Time to hold melons before pre-cooling/cooling should be minimised to prevent microbial growth.Provision of sanitary stations and field toilets with water, soap, and single-use towels, as well as their usage by staff, should be ensured.Food safety and risk management training for the staff.Post-harvest risk management examples:Dry dumping is recommended over wet dumping.In case wet dumping is used, the water should be of drinking water quality, disinfected, with sanitiser levels, and the fruit and water temperature should be regularly monitored.Single-use water should be preferred; if recycled water is used, changes should be monitored by objective measurements.Harvested melons should be pre-cooled to 5–8 °C.

Further details of the recommendations and specific information about irrigation/agriculture water quality, use of soil amendments, and post-harvest water quality criteria are available in Bartlett et al. [7] and FDA [52].

## 6. Conclusions

Rockmelons (cantaloupes) have been implicated in many foodborne outbreaks in recent years. Since 2006, the majority of these disease outbreaks were attributed to *L. monocytogenes* and *Salmonella* contamination. In most cases, the fruit was contaminated at the farm, and the contaminating pathogen was transferred to the flesh during processing; later on, storage conditions or temperature abuse increased the bacterial population. A preventative approach is considered an effective strategy to mitigate food safety risks by reducing microbial hazards in rockmelons. Several guidelines and recommendation documents outline the best practices and approaches to ensure that pathogen reductions are achieved from production to consumption. Effectively operating GAPs and HACCP systems and implementing specifically designed strategies would reduce the risk of rockmelon contamination. For example, educating all personnel involved about the potential sources, routes of contamination, and control measures to reduce, control, or eliminate microbial contamination in rockmelons. Prohibiting the use of treated composts containing animal manure or poultry litter at the farm is also an effective way to minimise the risk of microbial cross-contamination. After harvesting, immediately transferring the rockmelons to the packhouse for pre-cooling, sorting, washing, sanitising, and packing is important to prevent cross-contamination and microbial growth. Furthermore, an environmental monitoring program is needed to validate cleaning and sanitising protocols for all equipment, machinery, and packhouses. It is crucial to manage the food safety risks associated with the production and consumption of rockmelons to prevent future disease outbreaks and protect public health.

## Figures and Tables

**Figure 1 foods-13-02198-f001:**
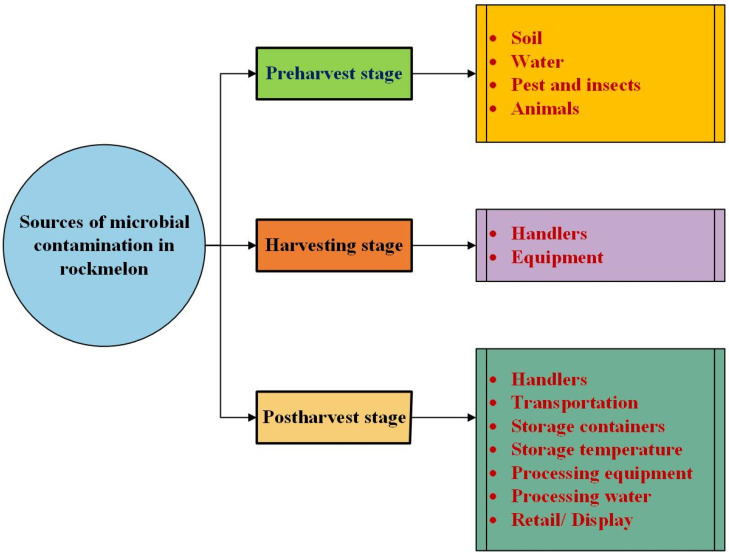
Common sources of microbial contamination in rockmelons at different supply chain stages.

**Table 1 foods-13-02198-t001:** Outbreaks associated with consumption of contaminated rockmelons.

Year	Country	Foodborne Pathogens Involved	Cases	Hospitalisation	Deaths	Fatality Rate (%)	References
2023	USA	*Salmonella* Sundsvall and Oranienburg	407	158	6	1.5%	[23]
2023	Canada	*S.* Soahanina, S. Sundsvall, *S.* Oranienburg, and *S.* Newport	190	68	9	4.7%	[24]
2022	USA	*S.* Typhimurium	87	32	0	N/A	[25]
2018	Australia	*L. monocytogenes*	22	N/A *	8	36.4%	[7,21]
2016	Australia	*S.* Hvittingfoss	144	N/A	0	N/A	[22]
2012	USA	*S.* Typhimurium and *S.* Newport	261	94	3	1.1%	[26]
2011	USA	*L. monocytogenes*	147	143	33	22.4%	[9]
2011	USA	*S.* Panama	20	3	0	N/A	[27]
2006	Australia	*S.* Saintpaul	100	9	0	N/A	[17]

* N/A means not applicable.

**Table 2 foods-13-02198-t002:** Recalls associated with microbiological contamination of rockmelons.

Foodborne Pathogens	Year	Country	Reference
*Salmonella* Sundsvall and *S.* Oranienburg	2023	USA	[23]
*S.* Soahanina, *S*. Sundsvall, *S.* Oranienburg, and *S.* Newport	2023	Canada	[24]
*L. monocytogenes*	2018	Australia	[7,21]
*S.* Hvittingfoss	2016	Australia	[28]
*S.* Typhimurium and *S.* Newport	2012	USA	[27]
*L. monocytogenes*	2011	USA	[9]

**Table 3 foods-13-02198-t003:** List of selected studies reporting the prevalence of microbiological hazards in rockmelons.

Foodborne Pathogen	Year	Country	Prevalence	Sample Type	Location	References
*E. coli*	2000	USA	950 (37, 3.9%)	Surface samples collected at the farm	Field	[39]
	2000	Mexico	300 (77, 25.7%)	Surface samples collected at the farm	Field	[39]
	2012–2016	Canada	ND	Fresh-cut rockmelon	Retail	[40]
	2020–2021	Portugal	ND *	Fresh-cut rockmelon	Retail	[41]
	2023	Portugal	------	Cut surface of rockmelon and rockmelon peel	Retail	[42]
*L. monocytogenes*	2012–2016	Canada	699 (5, 0.72%)	Fresh-cut rockmelon	Retail	[40]
	2009–2013	Canada	140 (2, 1.4%)	Fresh-cut rockmelon	Retail	[43]
	2012–2014	Germany	ND	The rind of reticulated rockmelons	Unreported	[10]
	2015	USA	16 (1, 6.3%)	Rinse samples from whole rockmelons.	Farmer’s Markets	[44]
*Salmonella*	2014–2015	Germany	147 (3, 2.1%)	Rind and pulp samples from peeled rockmelon	Unreported	[10]
	2009–2013	Canada	2400 (2, 0.08%)	Outside of whole rockmelons	Retail	[43]
	2000	USA	250 (2, 0.8%)	Surface swabs of whole rockmelon	Farms	[39]
	2000	Mexico	ND	Surface swabs of packed whole rockmelons	Farms	[39]
	2009–2014	USA	1075 (2, 0.19%)	Peel samples from the whole rockmelon	Retail	[40,45]
	2015	USA	16 (9, 56.3%)	Rinse samples from whole rockmelons	Farmer’s Markets	[44]

* ND means not detected.

## Data Availability

No new data were created or analyzed in this study. Data sharing is not applicable to this article.

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
