# Peer review of "Microbial Contamination and Disease Outbreaks Associated with Rockmelons (Cucumis melo): Implications for Public Health Protection"

_foods, 2024, doi:10.3390/foods13142198_

Round 1

Reviewer 1 Report

Comments and Suggestions for Authors

Attached 

Comments on the Quality of English Language

Attached

Author Response

Response: We thank the reviewer’s comments and suggestions to improve the manuscript. We have taken the comments positively and addressed them in the revised manuscript.

Below are the major comments:

1) Truncate the abstract and highlight key parameters such as methodology and technological advancements in detecting foodborne pathogens.

Response: The abstract has been modified by including more information.

2) What is the significance of Listeria monocytogenes, E. coli, and Salmonella spp in a microbiological study?

Response: These are the main foodborne pathogens in relation to rockmelon safety. Importance is given in lanes 112-113  and lanes 124-130.  

3) The introduction should be more widespread and should include a detailed discussion on foodborne pathogens, and their detection methods for continuous monitoring and diagnosis.

Response: The introduction has been updated by adding additional information.

4) Authors should consider adding these relevant references in the manuscript: 10.3389/fmicb.2021.710085; doi.org/10.3390/bios13020246

Response: Suggested references have been added to the manuscript.

5) Are there any POC test kits available to detection foodborne pathogens or viruses? What is the impact of these microsystems?

Response: Detection of foodborne pathogens is not the main focus of this manuscript, Information about monitoring the pathogens is given in section 5.

6) Among all the pathogens which is the most severe pathogen that needs to be resolved.

Response: yes, deaths were caused by Listeria infections (see lanes 110-130 and Table 1).

7) Improve Fig. 1 as this shows minimal information.

Response: The purpose of Fig. 1 is to highlight only common sources of microbial contamination in rockmelons without making it a complicated graphical presentation. We have included a graphical abstract to further elaborate the contamination routes in rockmelons.

8) Add more literature figures that need rigorous discussion.

 Response: Additional literature has been used to discuss contamination and disease outbreaks.

9) Add a comparison table including 2-3 recent state-of-the-art showcasing LOD, dynamic range, and sensitivity of the devices.

Response: This is not the focus of this manuscript.

10) What are the challenges and limitations of this case study?

11) All the references should be rephrased and rechecked with recent state-of-the-art.

Response: All references for in-text citations and list at the end have been cross-checked to improve formatting.

12) Improve the English language.

 Response: The manuscript has been thoroughly edited to improve the English language.

Reviewer 2 Report

Comments and Suggestions for Authors

The literature review submitted by Rabiee et al. describes the occurrence of three significant pathogens in the melon production chain, attempting to present information on foodborne outbreaks, the epidemiology of contamination, and potential preventive measures. However, upon reading the review, it feels as though it fails to make any substantial contributions to the field of study. The ideas are presented superficially, lacking depth in the topics discussed, and the organization is confusing to read. Numerous times, the review repeats concepts and ideas that are already well-established in the field of food safety.

The attempt to summarize the impacts and contamination pathways of these foods, as well as to establish/suggest prophylactic measures, was not fully achieved in the review. Additionally, it is evident that few of the references used are current.

Remarks:

Use "Salmonella spp." in the title.

In the keywords, avoid repeating words already present in the title for better indexing. Replace them with words that contextualize the study area.

Lines 38-41 – Scientific names should appear in full, without abbreviations, the first time they are used in the manuscript. From that point on, use the genus abbreviation. Double-check the use of italics for all scientific names in the manuscript.

Line 56 – Listeria genus or L. monocytogenes? You must be specific and precise throughout the manuscript.

Line 64 – Revise the sentence construction, as it appears fragmented and nonsensical.

Line 67 – Species or serovars? There are only two species of Salmonella, and most outbreaks are attributed to S. enterica.

In Table 1, the taxonomy of Salmonella needs to be reviewed. The serovars are written as if they were species. Use capital letters without italics for the serovars.

In Table 2 and the following text, the authors should show the economic impact of the recalls.

Figure 1 – What do the authors mean by "modes"?

Figure 1 is vague and unspecific, pointing not only to contamination sources but also to stages in the food production chain.

Lines 103-122 – Are all these pieces of information supported solely by reference number 21?

Lines 124-130 – Include a reference.

Topic 4 should be subdivided for each microorganism. Each paragraph discusses a different microorganism in an interspersed manner, making reading and comprehension difficult. This topic requires restructuring.

Lines 244-254 – The authors need to include bibliographic references for all their assertions in the literature review. Several paragraphs are presented without the necessary references.

The conclusion is not a true conclusion. It is merely a repetition of information.

Author Response

Response: We thank the reviewer’s comments and suggestions to improve the manuscript. We have taken the comments positively and addressed them in the revised manuscript.

Remarks:

Use "Salmonella spp." in the title.

Response: The title has been changed to align with other reviewer’s suggestions.

In the keywords, avoid repeating words already present in the title for better indexing. Replace them with words that contextualize the study area.

Response: Keywords have been revised.

Lines 38-41 – Scientific names should appear in full, without abbreviations, the first time they are used in the manuscript. From that point on, use the genus abbreviation. Double-check the use of italics for all scientific names in the manuscript.

Response: – Scientific names, use of the genus, abbreviations, and italics have been thoroughly checked in the revised manuscript.

Line 56 – Listeria genus or L. monocytogenes? You must be specific and precise throughout the manuscript.

Response: Suggested change has been made.

Line 64 – Revise the sentence construction, as it appears fragmented and nonsensical.

Response: Suggested change has been made.

Line 67 – Species or serovars? There are only two species of Salmonella, and most outbreaks are attributed to S. enterica.

 Response: Suggested changes have been made.

In Table 1, the taxonomy of Salmonella needs to be reviewed. The serovars are written as if they were species. Use capital letters without italics for the serovars.

Response: Table 1 has been updated

In Table 2 and the following text, the authors should show the economic impact of the recalls.

Figure 1 – What do the authors mean by "modes"?

Response: Figure 1 has been updated

Figure 1 is vague and unspecific, pointing not only to contamination sources but also to stages in the food production chain.

Response: The purpose of Fig. 1 is to highlight only common sources of microbial contamination in rockmelons without making it a complicated graphical presentation. We have included a graphical abstract to further elaborate the contamination routes in rockmelons.

Lines 103-122 – Are all these pieces of information supported solely by reference number 21?

 Response: New references have been added.

Lines 124-130 – Include a reference.

Response: New references have been added.

Topic 4 should be subdivided for each microorganism. Each paragraph discusses a different microorganism in an interspersed manner, making reading and comprehension difficult. This topic requires restructuring.

 Response: Suggested changes have been made.

Lines 244-254 – The authors need to include bibliographic references for all their assertions in the literature review. Several paragraphs are presented without the necessary references.

The conclusion is not a true conclusion. It is merely a repetition of information.

Response: The conclusion has been modified.

Reviewer 3 Report

Comments and Suggestions for Authors

The authors have drafted an interesting review paper but I feel that some parts of the review are misplaced which makes the draft hard to follow. Further, they should try as much as possible to position in the review in the global context, which would eventually spark a global interest. My suggestions are as follows;

-The title should be rephrased to read ‘‘Microbial contamination and disease outbreaks associated with rockmelons (Cucumis melo): Implications for public health protection’’

-The abstract is very shallow as no mention is made of the aim of the review, the approach used to retrieve literature, major findings, future research directions were made. For example, the authors did not mention any incidences of illnesses or major disease outbreaks due to consumption of microbially contaminated rockmelons to date.

-For keywords, do not repeat words already in the title.

-Other major suggestions are in the attached manuscript file.

Comments on the Quality of English Language

Needs to be rechecked throughout 

Author Response

Response: We thank the reviewer’s comments and suggestions to improve the manuscript. We have taken the comments positively and addressed them in the revised manuscript.

My suggestions are as follows;

-The title should be rephrased to read ‘‘Microbial contamination and disease outbreaks associated with rockmelons (Cucumis melo): Implications for public health protection’’

Response: We have accepted the suggested title and made changes to the manuscript.

-The abstract is very shallow as no mention is made of the aim of the review, the approach used to retrieve literature, major findings, future research directions were made. For example, the authors did not mention any incidences of illnesses or major disease outbreaks due to the consumption of microbially contaminated rockmelons to date.

Response: The abstract has been rewritten to include the suggested information.

-For keywords, do not repeat words already in the title.

Response: The keywords have been updated.

-Other major suggestions are in the attached manuscript file.

Response: We have included all suggested changes in the revised manuscript.

Round 2

Reviewer 1 Report

Comments and Suggestions for Authors

Authors have made significant changes to the revised manuscript and now the revised version looks good and can be accepted for publication in its current form.

Author Response

There are no specific comments required.

Reviewer 2 Report

Comments and Suggestions for Authors

Dear Editors,

In the previous review, I had suggested rejection due to the lack of novelty and insufficient merit for publication. As the editors decided to proceed with the evaluation of the manuscript despite my important consideration on this matter, I do not have any new comments to make, as all my minor remarks were addressed during the first round of revisions.

Author Response

(The authors gave the same response as above.)

Reviewer 3 Report

Comments and Suggestions for Authors

Most of my concerns on the manuscript have been addressed. There's something to correct: in Figure 1, wild animals and insects should be put under pests (since pests include even insects, other animals, viruses or even bacteria). It is also not clear how human activities are different from stuff like retail, transportation etc. The figure needs to be rethought about.

Comments on the Quality of English Language

Minor fixes required

Author Response

Figure 1 has been updated.